# SAMHD1 Attenuates Acute Inflammation by Maintaining Mitochondrial Function in Macrophages via Interaction with VDAC1

**DOI:** 10.3390/ijms24097888

**Published:** 2023-04-26

**Authors:** Bowen Xu, Qianyi Sui, Han Hu, Xiangjia Hu, Xuchang Zhou, Cheng Qian, Nan Li

**Affiliations:** 1National Key Laboratory of Immunity & Inflammation, Institute of Immunology, Naval Medical University, Shanghai 200433, China; 2Faculty of Anesthesiology, Changhai Hospital, Naval Medical University, Shanghai 200433, China; 3School of Sport Medicine and Rehabilitation, Beijing Sport University, Beijing 100084, China

**Keywords:** SAMHD1, sepsis, macrophage, mitochondrion, VDAC1, M1 polarization, innate immunity, molecular mechanism, TLR4 signaling, acute inflammation

## Abstract

Over-activation of Toll-like receptor 4 (TLR4) is the key mechanism in Gram-negative bacterial infection-induced sepsis. SAM and HD domain-containing deoxynucleoside triphosphate triphosphohydrolase 1 (SAMHD1) inhibits multiple viruses, but whether it plays a role during bacterial invasion remains unelucidated. Monocyte-macrophage specific *Samhd1* knockout (*Samhd1^−/−^*) mice and *Samhd1^−/−^* macrophage cell line RAW264.7 were constructed and used as research models to evaluate the role of SAMHD1 in TLR4-activated inflammation. In vivo, LPS-challenged *Samhd1^−/−^* mice showed higher serum inflammatory factors, accompanied with more severe inflammation infiltration and lower survival rate. In vitro, *Samhd1^−/−^* peritoneal macrophages had more activated TLR4 pathway upon LPS-stimulation, accompanied with mitochondrial depolarization and dysfunction and a higher tendency to be M1-polarized. These results could be rescued by overexpressing full-length wild-type SAMHD1 or its phospho-mimetic T634D mutant into *Samhd1^−/−^* RAW264.7 cells, whereas the mutants, dNTP hydrolase-function-deprived H238A and phospho-ablative T634A, did not exert the same effect. Lastly, co-IP and immunofluorescence assays confirmed that SAMHD1 interacted with an outer mitochondrial membrane-localized protein, voltage-dependent anion channel-1 (VDAC1). SAMHD1 inhibits TLR4-induced acute inflammation and M1 polarization of macrophages by interacting with VDAC1 and maintaining mitochondria function, which outlines a novel regulatory mechanism of TLR signaling upon LPS stimulation.

## 1. Introduction

Over-activation of Toll-like receptor 4 (TLR4) is the key molecular pathology of Gram-negative bacterial infection-induced sepsis, in which regulation of mitochondria remains elusive. Lipopolysaccharide (LPS)-activated inflammatory macrophages (M1) constitute the first line of host defense against invading Gram-negative bacteria [1]. Specifically, activated M1 macrophages produce abundant pro-inflammatory mediators, including IL-6, TNFα, IL-1, nitric oxide (NO), and reactive oxygen species (ROS), to kill the invading pathogens [2]. Nevertheless, if M1-mediated inflammation exerts excessive intensity, a cytokine storm can develop, thus causing the occurrence of severe sepsis [3]. Conversely, M2 macrophages express anti-inflammatory cytokines, such as IL-10, thus inducing an LPS-tolerant state in the host as the infection persists [4]. The metabolic state influenced by mitochondrial function plays a key role in macrophage M1/M2 polarization. Specifically, M1 macrophages block tricarboxylic acid cycle (TAC) and oxidative phosphorylation. They activate glycolysis and pentose phosphate metabolism as a substitute. M2 macrophages have intact TAC and oxidative phosphorylation. They oxidate pyruvate and fatty acid to produce adenosine triphosphate (ATP) [5,6]. In addition, a prior study revealed that mitochondrial dysfunction could impede the reprogramming of M1 macrophages to M2 macrophages, whereas M2 macrophages, which have intact mitochondria, were more readily repolarized to M1 macrophages upon LPS and IFNγ induction [7].

SAM and HD domain-containing deoxynucleoside triphosphate triphosphohydrolase 1 (SAMHD1), a regulatory molecule of the innate immune system found in a wide range of cells, is especially highly expressed in dendritic cells and macrophages (including microglia, Kupffer cells, Langerhans cells, and other macrophages localized in specific tissues). SAMHD1 is a deoxy-ribonucleoside triphosphate hydrolase (dNTPase), whose activity is responsible for hydrolysis of dNTPs into deoxynucleosides (dN) and inorganic triphosphates (PPPi) [8]. Accordingly, this molecule is critical for regulating deoxy-ribonucleoside triphosphate (dNTP) metabolism and maintaining homeostasis of the intracellular dNTP pool [9]. Furthermore, proper quantity and balance of the dNTP pool plays a pivotal role in DNA replication and genome stability. For example, an insufficient dNTP pool results in replication stress [10], while excessive or disturbed dNTP pools may lead to mismatched base pairing and DNA polymerase stall during replication [11,12]. As reported, *Samhd1* deficiency can contribute to dNTP pool imbalance and genomic instability, which may be associated with elevation in IFN-α level and the occurrence of Aicardi–Goutières syndrome [13]. In non-proliferating cells (such as resting CD4^+^ T cells and macrophages), SAMHD1 blocks the reverse transcription of viruses by decreasing the intracellular dNTP concentration [14]. Previous studies have unveiled that SAMHD1 has the ability to repress the replication of many RNA and DNA viruses such as human immunodeficiency virus 1 (HIV-1), equine infectious anemia (EIA) virus, and human papillomavirus virus 16 (HPV-16) [15,16,17]. Additionally, SAMHD1 can maintain genome stability by recruiting Mre11, a DNA double-strand break repairing protein, to stalled replication forks [18], or by reducing the activity of long interspersed element 1 (LINE-1) retrotransposons through multiple pathways independent of dNTPase [19,20,21].

Our preliminary experiments showed that SAMHD1 expression was significantly elevated in macrophages upon LPS stimulation. Therefore, we hypothesized that SAMHD1 was functional in this process. Through in vivo and in vitro LPS-stimulated models of *Samhd1^−/−^* mice and macrophages, we found that SAMHD1 was able to decrease TLR4 signaling through maintaining the mitochondrial state, which was dependent on its dNTPase function and regulated by the phosphorylation site T634. Deficiency of *Samhd1* led to acute inflammation and M1 polarization of macrophages. We also proved that SAMHD1 interacted with voltage-dependent anion channel-1 (VDAC1), which happens to be an outer mitochondrial membrane (OMM)-localized protein that is vital to mitochondria, and deficiency of *Samhd1* resulted in down-regulation of VDAC1. This indicated that SAMHD1 may exert its mitochondria-regulating function through VDAC1.

## 2. Results

### 2.1. Myeloid Samhd1 Knockout Exacerbated LPS-Induced Sepsis in Mice

To investigate whether SAMHD1 expression was related to TLR4 activation, peritoneal macrophages from wild-type mice were stimulated with LPS for 3 h. The results of qPCR showed that the mRNA expression of *Samhd1* was significantly elevated in macrophages upon LPS stimulation. Moreover, western blotting results revealed that the protein expression of SAMHD1 was increased sequentially in mouse peritoneal macrophages upon LPS stimulation (Figure 1A). These results suggested that SAMHD1 may be involved in the pathological process of acute inflammation. Accordingly, we bred myeloid *Samhd1* knockout (*Samhd1^−/−^*) mice to evaluate the function of SAMHD1 in vivo and in vitro. The levels of IL-6, TNFα, and lactate were significantly enhanced in the serum of *Samhd1^−/−^* mice after 3 h intraperitoneal injection of LPS (15 mg/kg) compared to that of control mice (Figure 1B). Furthermore, the lung tissues were collected for preparing hematoxylin and eosin (H&E)-stained sections. There was no marked change in the tissue morphology of *Samhd1^−/−^* or control mice after PBS injection. However, *Samhd1^−/−^* mice had thickened alveolar walls, increased inflammatory cell infiltration, and smaller alveolar lumen after LPS injection compared to control mice (Figure 1C). Next, *Samhd1^−/−^* and control mice were challenged with lethal doses of LPS (25 mg/kg) intraperitoneally. The survival rate of *Samhd1^−/−^* mice was significantly lower than that of control mice (Figure 1D). These findings indicated that myeloid *Samhd1* deficiency increased acute inflammation of LPS-induced sepsis.

### 2.2. Samhd1 Knockout Increased TLR4 Signaling in Macrophages

To confirm the negative role of SAMHD1 in TLR4-induced inflammation in vitro, we tested the inflammatory factor in macrophages. The mRNA expressions of *Il-6*, *Tnfα* and *Ifnβ* were significantly higher in *Samhd1^−/−^* macrophages than in that of control cells (Figure 2A). ELISA results also showed that the protein levels of IL-6, TNFα, and IFNβ in the supernatants of *Samhd1^−/−^* macrophages were higher than in control macrophages (Figure 2B). The above results are consistent with the results of in vivo experiments, so we speculated that the TLR4 pathway might have a superior activation in *Samhd1^−/−^* macrophages than control cells upon LPS stimulation. The LPS-TLR4 pathway consists of two main pathways, including MyD88-dependent activation of nuclear factor kappa-B (NF-κB) or activator protein 1 (AP-1) which induce the expression of proinflammatory cytokine genes, and MyD88-independent activation of interferon regulatory factor 3 (IRF3), NF-κB, and AP-1 which induce type I interferons [22]. Therefore, western blotting was conducted to test the expression and phosphorylation of key proteins in these pathways upon LPS stimulation in macrophages of *Samhd1^−/−^* or control mice. Although the total amount of these key proteins did not differ markedly, the phosphorylation levels were generally higher in *Samhd1^−/−^* macrophages than in control macrophages (Figure 2C), confirming that *Samhd1* deficiency was associated with extensive activation of the LPS-TLR4 pathway in macrophages. 

### 2.3. Samhd1 Deficiency Enhanced M1 Differentiation

A prior study reported that generally, activation of the NF-κB pathway promoted macrophage M1 polarization, whereas inhibition of the NF-κB pathway accelerated macrophage M2 polarization [23]. Therefore, considering the above results that *Samhd1* deficiency promotes TLR4-NF-κB pathway activation in macrophages after LPS stimulation, we hypothesized that *Samhd1* deficiency might also lead to hyperpolarization of macrophages toward the M1 phenotype. In vivo, *Samhd1^−/−^* and control mice were given LPS intraperitoneally (15 mg/kg) or injected with PBS as sham operation then the M1 and M2 polarization of lung macrophages was detected using flow cytometer after 3 h. As expected, the proportion of M1-polarized lung macrophages was markedly higher in *Samhd1^−/−^* mice than that of control mice. Interestingly, even in PBS-treated mice, the proportion of M1-polarized lung macrophages in *Samhd1^−/−^* mice remained significantly higher than that of control mice, suggesting *Samhd1^−/−^* macrophages could differentiate spontaneously toward M1 phenotype even under normal conditions (Figure 3A). In vitro, peritoneal macrophages were induced to M1 or M2 phenotype. After 24 h, *Samhd1^−/−^* macrophages showed an increased proportion of M1 macrophages upon M1 induction and a decreased proportion of M2 macrophages upon M2 induction, compared with control cells, respectively (Figure 3B). Thus, the deficiency of *Samhd1* facilitated M1-polarization of macrophages.

### 2.4. Samhd1 Deficiency Impaired Mitochondrial Membrane Potential and Function in Macrophages but Did Not Affect Apoptosis

Macrophage M1 polarization is usually accompanied by an energy metabolic switch, namely inhibition of the mitochondrial electron transport chain and enhancement of the glycolytic pathway [6]. Therefore, we further probed the difference in the state of mitochondria between *Samhd1^−/−^* and control macrophages. The macrophages were induced toward M1 phenotype for 12 h, then JC-1 (a mitochondrial membrane potential probe) was loaded to cells and detected with a flow cytometer or fluorescence plate reader. *Samhd1^−/−^* macrophages had a higher proportion of green monomeric JC-1 than control macrophages, both at the resting state and after M1 polarization induction (Figure 4A). JC-1 is specifically enriched in mitochondria in normal conditions, but if mitochondrial membrane potential is low, JC-1 will leak into cytoplasm and meanwhile change its emission color from red to green. Hence, our results revealed lower mitochondrial membrane potential in *Samhd1^−/−^* macrophages. Subsequently, the fluorescence intensity of cells was detected separately in red (an excitation (Ex) wavelength of 525 ± 10 nm and an emission (Em) wavelength of 590 ± 10 nm) and green (an Ex wavelength of 490 ± 10 nm and an Em wavelength of 535 ± 10 nm) channels with a fluorescence plate reader. The ratio of red to green fluorescence intensity of *Samhd1^−/−^* macrophages was significantly lower than that of control macrophages at both resting state and M1 polarized state, illustrating mitochondrial impairment (Figure 4B). Mitochondrial stress assay demonstrated that the mitochondrial function of *Samhd1^−/−^* macrophages was inhibited compared to control macrophages at resting state. Glycolysis stress assay showed the glycolysis level of *Samhd1^−/−^* macrophages was higher than that of control macrophages at both the resting state and M1 polarized state, which might be a result of energy compensation (Figure 4D). Afterward, the apoptosis of *Samhd1^−/−^* and control macrophages was compared in resting, M1 polarized, and M2 polarized states using the Annexin-V Apoptosis Kit, which revealed no significant apoptosis in any of the three states (Figure 4C).

### 2.5. The Inhibition of Inflammation and Protection of Mitochondria Was Dependent on the dNTPase Function of SAMHD1 and Regulated by the Phosphorylation Site T634

To further clarify how *Samhd1* deficiency caused the above results, we constructed *Samhd1^−/−^* RAW264.7 (a mouse macrophage cell line). The protein levels of IL-6 and TNFα in cell supernatants were tested by ELISA after 12 h of LPS stimulation (100 ng/mL). Significantly higher concentrations of IL-6 and TNFα were exhibited in *Samhd1^−/−^* RAW264.7 cells than in control cells. This result is concurrent with the result in *Samhd1^−/−^* peritoneal macrophages (Figure 5A). Existing results show many functions of SAMHD1 are exerted through its dNTPase activity, and H238A/D239A double mutation of mouse SAMHD1 (the following is omitted to be H238A) is known to cause loss of dNTPase function [24]. Moreover, a phosphorylation site on human SAMHD1, T592, is phosphorylated by cyclin dependent kinase (CDK)1 and CDK2, thereby reducing resistance to HIV-1 viruses [25]. T634 in mouse SAMHD1 was validated to be a homologous site of T592 in human SAMHD1, which could be phosphorylated by mouse CDK1 and CDK2 [24]. Consequently, we constructed eukaryotic overexpression vectors of full-length mouse SAMHD1, dNTPase-function-deprived H238A mutant, phospho-ablative alanine residue T634A mutant, and phospho-mimetic aspartic acid residue T634D mutant. *Samhd1^−/−^* RAW264.7 cells were electro-transfected with the above vectors, respectively, and then stimulated with LPS (100 ng/mL) for 12 h. Cell culture supernatants were collected for detection of cytokine secretion by ELISA and intracellular mitochondrial membrane potential of cells was tested by JC-1. As a result, compared with cells overexpressing vector, IL-6 levels in the supernatants were markedly lower when cells overexpressed full-length SAMHD1 or T634D mutant, but higher when cells overexpressed H238A mutant. There is no significant difference in IL-6 levels of supernatants between cells overexpressing vector and T634A mutant (Figure 5B). The ratio of JC-1 red to green fluorescence intensity was substantially higher in cells overexpressing full-length SAMHD1 or T634D mutant but lower in cells overexpressing H238A mutant, compared with that of vector control. Meanwhile, no significant difference was observed in the ratio of JC-1 red to green fluorescence intensity between cells overexpressing T634A mutant and vector (Figure 5C). To test whether these mutations can affect M1 polarization, *Samhd1^−/−^* RAW264.7 cells were electro-transfected with the above vectors, respectively, and M1 polarization was induced for 24 h. Afterwards, the proportion of M1-polarized macrophages was tested with flow cytometry. The results revealed that the proportion of M1-polarized macrophages was lower in cells overexpressing full-length SAMHD1 but higher in cells overexpressing H238A than in control cells. In addition, T634A-overexpressing cells had a higher proportion of M1-polarized macrophages than T634D-overexpressing cells (Figure 5D).

### 2.6. SAMHD1 Regulates Mitochondrial Function via Interacting with VDAC1

To further investigate the molecular mechanism of how SAMHD1 regulates mitochondria, an IP-MS was conducted to screen its interacting molecules. The results revealed that VDAC1 was one of the potential candidates (Appendix A). VDAC1 and the other two molecules in its family, VDAC2 and VDAC3, are highly conserved in their structure and are redundant and complementary in their function. VDAC family is channel proteins mainly localized on OMM and is responsible for transmembrane material transport [26]. The interacting of SAMHD1 with VDAC1, but not with VDAC2, was confirmed by co-IP assay with RAW264.7 cells under normal conditions (Figure 6A). Western blotting results showed that *Samhd1* deficiency in macrophages decreased VDAC1 protein expression (Figure 6B). Co-localization of SAMHD1 and VDAC1 (indicated by white arrows) was observed under confocal microscope and negative SAMHD1 staining signals in *Samhd1^−/−^* macrophages indicated the satisfied specificity of the SAMHD1 antibody (Figure 6C). These results suggested SAMHD1 may promote mitochondrial function via interacting with VDAC1.

## 3. Discussion

The pathogenesis of sepsis has not been fully elucidated. The identification of LPS from bacterial cell walls by TLRs on the surface of macrophages generates a natural immune response, which is widely recognized as the basic model for the study of sepsis at present [27]. When macrophages are hyperpolarized towards M1 phenotype and anti-inflammatory M2 macrophages fail to balance the pro-inflammatory effects in the early stage of bacterial infection, it will promote uncontrolled local inflammation to systemic sepsis [28]. Energy metabolic pathways regulated by the functional state of mitochondrion are determinants of macrophage M1 or M2 polarization [5,6,29]. Therefore, mitochondrion is expected to be a key target for altering the course of sepsis.

It has been reported that SAMHD1 has dNTPase activity, which can hydrolyze dNTPs to dN and PPPi [8]. It is also implicated in the maintenance of intracellular dNTP pool homeostasis [9] and genomic stability [10,11,12,13] as well as antiviral activity [14]. However, little is known about whether SAMHD1 is involved in bacterial infection and sepsis. In our study, the results unveiled that SAMHD1 was upregulated in LPS-stimulated wild-type macrophages, suggesting that it may play a role in the pathological process of sepsis. In LPS-induced *Samhd1^−/−^* mice and *Samhd1^−/−^* peritoneal macrophage models, our data elucidated that *Samhd1* deficiency promoted the activation of the LPS-TLR4 pathway, triggering stronger systemic/local inflammation as well as higher lethality. *Samhd1^−/−^* peritoneal macrophages polarized to M1 more upon M1 induction and the situation is on the contrary upon M2 induction. We hypothesized that the difference in polarization between *Samhd1^−/−^* and control macrophages might be attributed to their different energetic metabolism. The JC-1 assay elaborated that the mitochondrial membrane potential of *Samhd1^−/−^* macrophages was lower than control macrophages both in resting and M1-polarized states. Meanwhile, mitochondrial stress assay and glycolysis stress assay demonstrated that the mitochondrial function of *Samhd1^−/−^* macrophages was impaired and compensated with glycolysis. Intriguingly, despite the damage in mitochondria of *Samhd1^−/−^* macrophages, there was no significant apoptosis of *Samhd1^−/−^* macrophages in M0, M1, and M2 states, indicating that mechanisms such as mitochondrial autophagy may exist in the cells to maintain their survival.

Subsequently, rescue experiments were conducted by overexpressing full-length mouse SAMHD1 in *Samhd1^−/−^* RAW264.7 cells. The results further clarified that SAMHD1 reduced secretion of inflammatory cytokines, maintained mitochondrial stability, and inhibited hyperpolarization toward the M1 phenotype in LPS-induced macrophages. Meanwhile, we also overexpressed several mutants of full-length SAMHD1 in *Samhd1^−/−^* RAW264.7 cells, among which the H238A mutant, which associated with the loss of dNTPase activity [24], could not reverse the above phenotypes, illustrating that SAMHD1′s inhibition of LPS-TLR4 inflammation and protection of mitochondria were dNTPase-dependent. Mouse SAMHD1 T634 has been identified as a phosphorylation site homologous to human SAMHD1 T592, the function of which is unclear. Although the phosphor-mimetic T634D mutant has been demonstrated to cause the loss of HIV-1 restriction in non-dividing cells, it does not affect murine leukemia virus infection in dividing cells [24]. Accordingly, we overexpressed the phospho-ablative alanine residue mutant T634A or the phospho-mimetic aspartic acid residue mutant T634D in *Samhd1^−/−^* RAW264.7 cells, respectively, and found that overexpression of the T634D mutant, but not the T634A mutant, exerted similar effects to overexpression of full-length SAMHD1. This result illustrates that the phosphorylation of T634 is a potential target for regulating the LPS-TLR4 inflammation. Of note, our rescue experiments were performed with immortalized RAW264.7 cells as a cell model. In this type of cell, CDK activity that regulates T634 phosphorylation may be different from that of terminally differentiated macrophages, which warrants further investigation.

Finally, VDAC1 was screened as the candidate molecule that interacted with SAMHD1 by IP-MS. Co-IP and immunofluorescent assay further verified that SAMHD1 could directly interact with VDAC1. Interestingly, VDAC1, which mainly localized on OMM, is the only known pathway of OMM that mediates the passage of ions, nucleotides, and other high-molecular-weight metabolites (such as pyruvate, malate, succinate, NADH/NAD as well as heme and cholesterol) [26]. In a former study, human SAMHD1 was found to be able to regulate mitochondrial dNTP pools, thus influenced the copy number of mitochondrial DNA in fibroblasts [30]. Though different in species and cell types, it is consistent with our research results that SAMHD1 is involved in mitochondrial maintenance in a dNTPase-dependent manner. We proved that *Samhd1* deficiency in macrophages could result in the down regulation of VDAC1 expression. Because VDAC1 is also a commonly recognized loading control of mitochondria for many quantitative assays [31], down regulation of VDAC1 means a reduction in total mitochondrial quantity, which is consistent with our results that mitochondria in *Samhd1^−/−^* macrophages are damaged. However, whether SAMHD1-VDAC1 interaction is indispensable and how it works in this process remains to be further investigated. Other research shows that SAMHD1 may inhibits activation of NF-κB during virus infection by directly binding to NF-κB1/2 and reducing phosphorylation of the NF-κB inhibitory protein IκBα [32] or through a negative regulation of TRAF6-TAK1 checkpoint [33]. SAMHD1 also inhibits IFN-I induction pathway by interacting with inhibitor-κB kinase ε (IKKε) and IFN regulatory factor 7 (IRF7) [32]. These results are consistent with ours (Figure 2), but also raise questions about which mechanism of SAMHD1 plays the leading role in its regulation of the TLR4 pathway. Though downregulation of mitochondrial function may be the result of TLR4 signaling in *Samhd1^−/−^* macrophages upon LPS stimulation, our results support the hypothesis that it is the mitochondrial dysfunction caused by *Samhd1* deficiency that leads to intensified TLR4 activation. We observed that impaired function of mitochondria already existed in *Samhd1^−/−^* macrophages before LPS was loaded (Figure 4). However, more work needs to be performed to further reveal how these mechanisms of SAMHD1 cooperate and interact with each other in regulation of TLR4 pathway.

In summary, our study unravels that SAMHD1 negatively regulates LPS-TLR4 pathway. SAMHD1 expression is elevated upon LPS-stimulation. *Samhd1* deficiency increases mitochondrial damage, which enhances LPS-TLR4 pathway and makes macrophages more M1 polarized, and SAMHD1-VDAC1 interaction may be involved in this process. Consistently, a higher level of inflammatory cytokine secretion aggravates local and systemic tissue damage and thus raises lethality in LPS-induced sepsis. Therefore, SAMHD1 is a potential target molecule against sepsis. In addition, we identified H238/D239 and T634 on mouse SAMHD1 as two key targets that regulate its anti-sepsis function.

## 4. Materials and Methods

### 4.1. Mouse Models of LPS-Induced Sepsis

All mice were housed under specific pathogen-free conditions and all experiments involving animals were performed in accordance with National Institute of Health Guide for the Care and Use of Laboratory Animals and approved by Scientific Investigation Board of Naval Medical University (Shanghai, China). The *Samhd1^−/−^* mice were generated using CRISPR-Cas9 technique by Cyagen Biosciences, Inc. (Suzhou, China). Specifically, the mouse *Samhd1*-specific gRNA (gRNA1: TGTTGACAGGAAGGGATCGCTGG; gRNA2: TTGTGGTGACCGTGAACTAAGGG), the donor vector containing loxP sites, and Cas9 mRNA were co-injected into fertilized mouse eggs for generating targeted conditional knockout offspring. Loxp sites are on both sides of exon 2 on *Samhd1*-encoding mRNA (NCBI Reference Sequence: NM_018851.4). F0 founder animals were identified by polymerase chain reaction (PCR) and then subjected to sequence analyses, which were mated to wild-type mice to test germline transmission and generate F1 animals. Target F1 mice were bred with tissue-specific lysozyme 2-*Cre*-deleted mice to generate F2 animals. Heterozygous *Cre^+^* mice were bred with homozygous mice to generate *Samhd1^−/−^* mice (male, aged 6–8 weeks; the experimental group) and *Samhd1^fl/fl^* mice (male, aged 6–8 weeks; the control group). Genomic DNA was isolated from tails and analyzed by PCR amplification with the use of the following primers: 5′-ACACTAGTAGTCCCTTCTGAGGTG-3′ and 5′-TCTTTACCACAATCTGCCTGACA-3′. LoxP homozygote had bands with 200 bp, wild-type had bands with 139 bp, and heterozygote had bands with both 200 bp and 139 bp.

Mice were injected intraperitoneally with LPS (15 mg/kg) for 3 h to establish a model of LPS-induced sepsis or with PBS as sham operation. Three hours after LPS injection, mice were anesthetized with isoflurane, and orbital blood was rapidly collected, followed by euthanasia of mice by cervical dislocation. Subsequently, the lung tissues were obtained and preserved in 4% paraformaldehyde solution, or ground into single cells using a syringe pusher in a 70 μm filter for subsequent flow cytometry or quantitative PCR (qPCR) assays. In the assay of observing survival rate of mice, the dose of LPS injection was 25 mg/kg. The observation lasted for 60 h, and we took completely incapacitated as the endpoint. Then, mice were euthanized by cervical dislocation after anesthesia.

### 4.2. Cell Culture

Peritoneal macrophages were isolated from mice aged 6–8 weeks as previously described [34]. RAW264.7 cells were obtained from American Type Culture Collection (ATCC, Manassas, VA, USA). All cells were cultured in a Dulbecco’s Modified Eagle Medium encompassing 10% fetal bovine serum (Gibco, Carlsbad, CA, USA) and 5% Penicillin-Streptomycin Solution (Sangon, Shanghai, China). Cells were stimulated with LPS (100 ng/mL) or treated with 20 ng/mL mIFNγ + 100 ng/mL LPS (M1 macrophage polarization) or with 20 ng/mL mIL-4 (M2 macrophage polarization) for indicated times.

CRISPR-Cas9-mediated ablation of *Samhd1* was achieved with CRISPR-Cas9 ribonucleoproteins (Haixing Bioscience, Zhaoqing, China) containing expression cassettes for hSpCas9 and chimeric guide RNA. To target exon 2 of *Samhd1*, two guide RNA sequences (gRNA1: CCAATCGGAATCCATTTGGGGGG; gRNA2: AAAGTTACCGCCCTCTTTGTAGG) were selected through a website (http://crispr.mit.edu. accessed on 12 July 2022). Plasmids containing the guide RNA sequences were electro-transfected into cells as instructed in the manuals of the Neon transfection system (Thermo Fisher, Waltham, MA, USA). After two days, single colonies were transferred into 96-well plates. To identify the presence of insertion or deletion (indels) in *Samhd1*-targeted clones, genomic DNA was isolated using a Quick-DNA Miniprep kit (Zymo Research, Orange, CA, USA), and PCR amplification was conducted using 2 × Taq Master Mix (Vazyme, Nanjing, China) of primers flanking exon (forward: 5′-AGCCATTTAGGGAGGGGTAGG-3′ and reverse: 5′-TGTGACCCAGGCAAGTTTCT-3′). Plasmids were isolated from 8–10 single colonies and sequenced with Sanger sequencing (GENEWIZ, Beijing, China). Clones with mutations in both alleles were selected for subsequent experiments.

### 4.3. RNA Quantification

Total RNA was collected with RNA extraction kits (Fastagen, Shanghai, China), 200 ng of which was subjected to reverse transcription with reverse transcription-PCR kits (Vazyme, Nanjing, China). qPCR was performed according to the protocols of Taq Pro Universal SYBR qPCR Master Mix (Vazyme, Nanjing, China). Data were normalized to β-actin expression. Relative changes in gene and mRNA expression were analyzed using the 2-ΔΔCt method. Primers are listed in Appendix A.

### 4.4. ELISA

IL-6, TNFα, and IFNβ in cell culture supernatants were measured using corresponding ELISA kits (R&D, Minneapolis, MN, USA). Blood was collected from the orbit of mice and stored at 4 °C overnight. After blood was centrifuged at 4 °C for 10 min, the supernatants were obtained as the specimen. Afterward, the levels of the cytokines were detected with corresponding ELISA kits (R&D, Minneapolis, MN, USA), and lactate levels were tested with an L-Lactate Assay Kit (Abcam, Cambridge, UK).

### 4.5. Immunoprecipitation (IP) and Immunoblot Analysis

Cells were lysed with cell lysis buffer, followed by the measurement of protein concentration with bicinchoninic acid kits (Thermo Fisher Scientific, Waltham, MA, USA). IP and immunoblot assays were performed as previously described [34]. Primary antibodies are presented in Appendix A.

### 4.6. Immunofluorescent Confocal Microscopy

After culturing, treating, fixing, blocking, permeating, and rinsing, cells were incubated on round glasses with primary antibodies (listed in Appendix A) and fluorochrome-conjugated secondary antibodies according to the manufacturer’s instructions. The round glasses were then loaded onto the slides with 4′,6-diamidino-2-phenylindole (DAPI)-containing mounting medium (Yeasen, Shanghai, China) and observed with a laser confocal microscope (Leica Microsystems, Wetzlar, German).

### 4.7. Flow Cytometry and Apoptosis Assay

Flow cytometry data were obtained from an ID7000 full-spectrum flow cytometer (Sony, Tokyo, Japan) and analyzed using the Flowjo software v10.8.1 (BD, Franklin Lakes, NJ, USA). Fluorescence signal channels are listed in Table 1. Gating strategy is shown in Appendix A. Single positive controls and fluorescence-minus-one (FMO) controls are listed in Appendix A, respectively.

Apoptosis was detected as directed in the manuals of Annexin V Apoptosis Kits (BD, Franklin Lakes, NJ, USA). In brief, cells were digested with trypsin, washed with PBS once and resuspended with Annexin V binding buffer containing FITC conjugated Annexin V antibody and PI. After incubating in the dark at room temperature for 15 min, cells were immediately resuspended using an appropriate amount of Annexin V binding buffer and detected by flow cytometer.

Polarization quantification of lung macrophages: Lung tissues were harvested, cut with scissors and digested with 100 U/mL type I collagenase (Absin, Shanghai, China) for 6 h at 37 °C. After filtering with 70 μm strainers, digestive residues were resuspended into single cells, among which lung macrophages were sorted by F4/80 positive selection magnetic beads (Stemcell, Vancouver, BC, Canada). Sorted macrophages were blocked with CD16/32 antibodies and then marked with conjugated antibodies including FITC-CD80, AF647-F4/80, BV421-CD11b, and PE-CD206, with the help of permeabilization reagents (Thermo Fisher Scientific, Waltham, MA, USA). The cells were then detected by flow cytometry. M1 and M2 macrophages were quantified among F4/80^+^ CD11b^+^ cells, according to the percentage of CD80^+^ cells and CD206^+^ cells, respectively.

Polarization quantification of peritoneal macrophages: Peritoneal macrophages were harvested, blocked with CD16/32 antibodies and then marked with conjugated antibodies including AF700-CD45.2, FITC-CD80, AF647-F4/80, BV421-CD11b, and PE-CD206, with the help of permeabilization reagents (Thermo Fisher Scientific, MA, USA). The cells were then detected by flow cytometry. M1 and M2 macrophages were quantified among CD45^+^ F4/80^+^ CD11b^+^ cells, according to the percentage of CD80^+^ cells and CD206^+^ cells, respectively.

Polarization quantification of RAW264.7 cell line: RAW264.7 cells were blocked with CD16/32 antibodies and then marked with conjugated antibodies including AF700-CD86, AF647-F4/80, and BV421-CD11b. The cells were then detected by flow cytometry. M1 macrophages were quantified among F4/80^+^ CD11b^+^ cells, according to the percentage of CD86^+^ cells.

### 4.8. Mitochondrial Membrane Potential Analysis

Macrophages were digested and incubated with 1X mitochondrial potential sensor JC-1 working solution (Yeasen, Shanghai, China) for 20 min in the dark at 37 °C. Then, macrophages were rinsed twice with JC-1 washing buffer as per the manufacturer’s instructions. The cells stained with JC-1 were loaded on a 96-well white-bottom plate. Signals were detected with a flow cytometer for flow cytometry diagrams or a fluorescence plate reader for quantification. Fluorescence signal channels of flow cytometry can be seen in Table 1. Channels of fluorescence plate reader: cell fluorescence intensity in red channel (an Ex wavelength of 525 ± 10 nm and an Em wavelength of 590 ± 10 nm) and green channel (an Ex wavelength of 490 ± 10 nm and an Em wavelength of 535 ± 10 nm) were detected. Ratio of red to green fluorescence intensity was used as an index of mitochondrial membrane potential.

### 4.9. Mitochondrial Respiration and Glycolysis Measurements

This assay was performed using an XF Extracellular Flux Analyzer (Agilent, Palo Alto, CA, USA) based on the protocols of Glycolytic Stress Test and Mitochondrial Test Kits.

### 4.10. Co-Immunoprecipitation Coupled to Mass Spectrometry (IP-MS)

IP was performed in the non-denatured lysates of RAW264.7 cells with SAMHD1 antibodies and protein A/G-binding agarose beads. Then, the precipitating complexes were boiled after addition of sodium dodecyl sulfate-polyacrylamide gel electrophoresis (SDS-PAGE) loading buffer, followed by electrophoresing on Bis-Tris Gels. When the lanes were about 1 cm long, electrophoresing was stopped and the lanes were cut off from the gels for mass spectrometry to identify the proteins in the lanes.

### 4.11. Plasmid Constructs and Transfection

The recombinant plasmids expressing full-length wild-type mouse SAMHD1 were constructed. Specifically, cDNA from RAW264.7 cells was subjected to PCR-based amplification and then subcloned into the pMAXCloning eukaryotic expression vector with a seamless cloning kit (Vazyme, Nanjing, China). Mutants, H238A, T634A, and T634D, were generated based on the full-length wild-type mouse SAMHD1 pMAXCloning vector with a quick mutation kit (Beyotime, Shanghai, China). The recombinant plasmids were amplified with competent E. coli DH5α and were purified again with an endotoxin-free large plasmid extraction kit (Abclonal, Wuhan, China). Plasmids were electro-transfected into cells using the 4D-Nucleofector System (Lonza, Basel, Switzerland) as per manufacturer’s instructions. Expression efficiency of the plasmids at the protein level was validated by Western blotting (Appendix A).

### 4.12. Statistical Analysis

All quantitative data were summarized as mean ± standard deviation. Comparisons between two groups were analyzed using two-tailed unpaired Student’s *t*-test. The survival rate was analyzed with Log-rank test. Differences with a *p*-value < 0.05 were considered statistically significant.

## Figures and Tables

**Figure 1 ijms-24-07888-f001:**
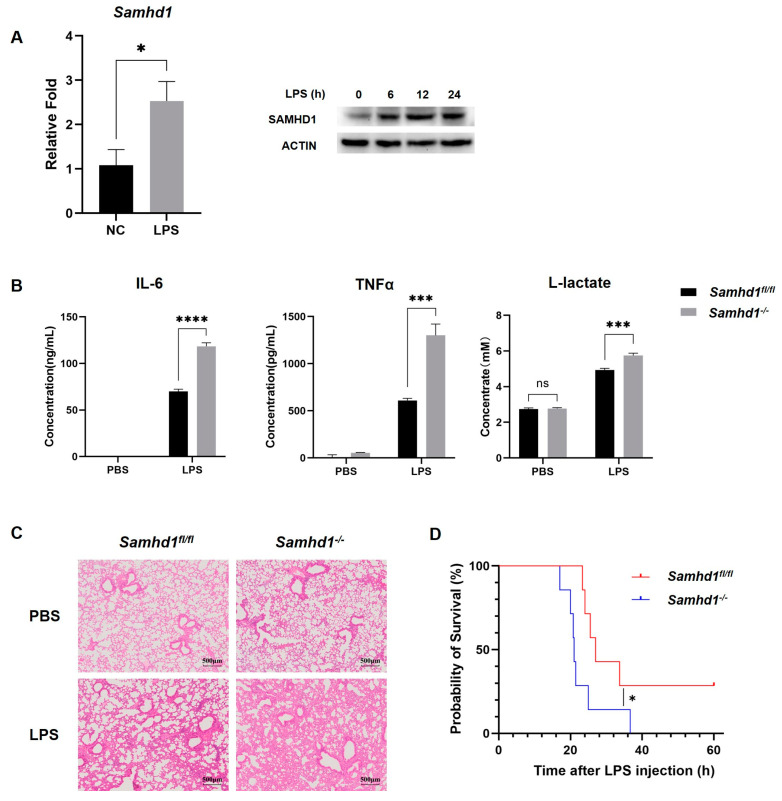
*Samhd1* deficiency increases acute inflammation in LPS-induced sepsis in vivo. (**A**) *Samhd1* mRNA expression was detected with qPCR in wild-type mice-derived peritoneal macrophages after 3 h LPS stimulation (100 ng/mL) (left) and SAMHD1 protein levels were detected with western blotting in mouse peritoneal macrophages that were stimulated with LPS for 6, 12, and 24 h (100 ng/mL) (right). * *p* < 0.05, Student’s *t*-test, *n* = 3/group. (**B**,**C**) *Samhd1^−/−^* and control mice were injected intraperitoneally with LPS (15 mg/kg) or PBS for 3 h. (**B**) The serum concentrations of IL-6, TNFα, and lactate were measured with ELISA and biochemical reagents, respectively, **** *p* < 0.0001, *** *p* < 0.001, Student’s *t*-test, *n* = 3/group, ns—not statistically significant. (**C**) Lung tissues of mice were collected for preparing H&E-stained sections. (**D**) *Samhd1^−/−^* mice or control mice were injected intraperitoneally with LPS (25 mg/kg), and the survival rate of mice was observed within 60 h. * *p* < 0.05, Log-rank test, *n* = 7/group.

**Figure 2 ijms-24-07888-f002:**
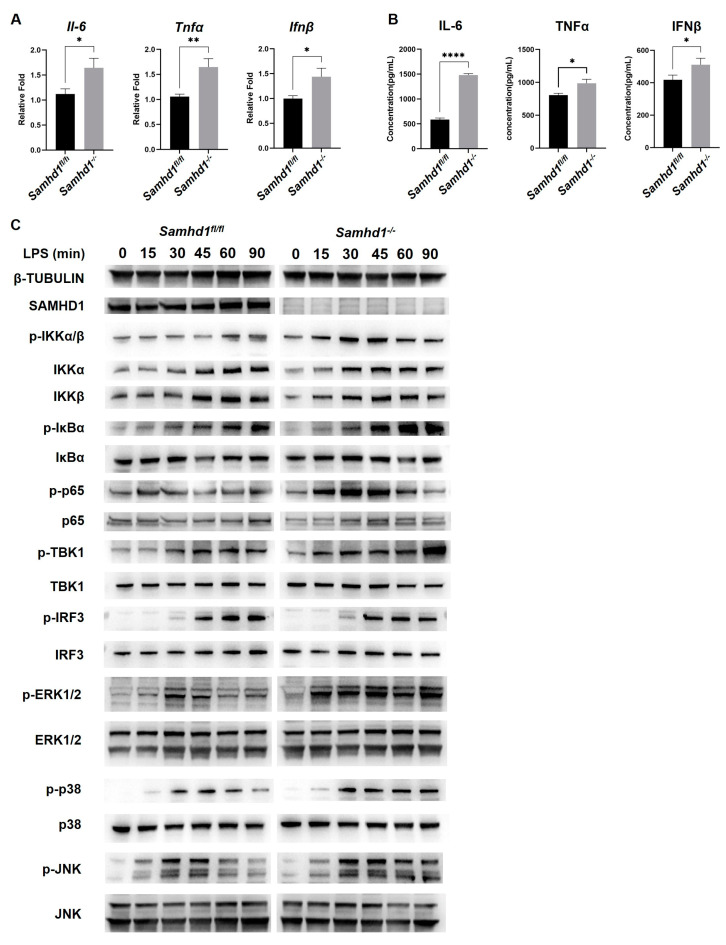
*Samhd1* deficiency increased TLR4 signaling and responses in vitro. (**A**) The mRNA expressions of *IL-6*, *TNFα*, and *IFNβ* were tested with qPCR in peritoneal macrophages from *Samhd1^−/−^* and control mice after 3 h LPS stimulation (100 ng/mL). * *p* < 0.05, ** *p* < 0.01, Student’s *t*-test, *n* = 3/group. (**B**) The protein expression of IL-6, TNFα, and IFNβ were detected with ELISA in cell supernatants of peritoneal macrophages from *Samhd1^−/−^* and control mice after 12 h LPS stimulation (100 ng/mL). * *p* < 0.05, **** *p* < 0.0001, Student’s *t*-test, *n* = 3/group. (**C**) The expression and phosphorylation of key proteins in TLR4 pathway were tested with western blotting in peritoneal macrophages from *Samhd1^−/−^* and control mice after 0–90 min of LPS stimulation (100 ng/mL).

**Figure 3 ijms-24-07888-f003:**
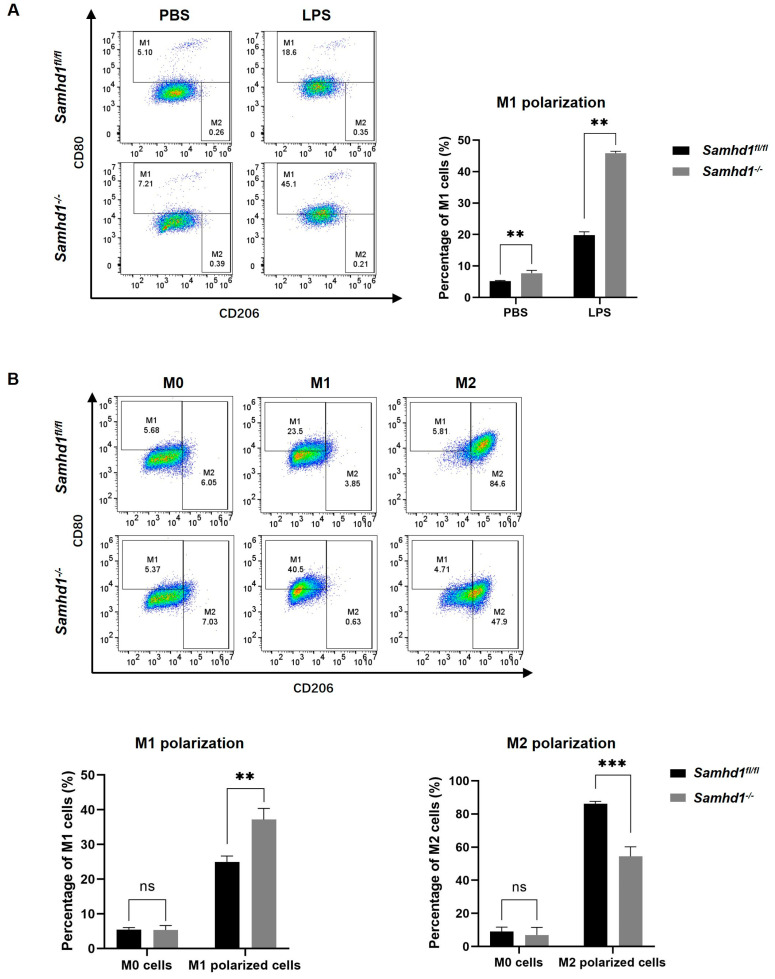
*Samhd1* deficiency induces M1 polarization. (**A**) Representative experiments showing that M1 and M2 polarization of lung macrophages from *Samhd1^−/−^* and control mice were detected with flow cytometry 3 h after intraperitoneal injection of LPS (15 mg/kg) or PBS. F4/80^+^ CD11b^+^ cells were gated and examined for CD80 and CD206 expression (left). Percentages of CD80^+^ (M1 phenotype) cells within total F4/80^+^ CD11b^+^ subsets are presented as bar graphs (right). ** *p* < 0.01, Student’s *t*-test, *n* = 3/group. (**B**) Representative experiments showing that M1 and M2 polarization of *Samhd1^−/−^* and control macrophages was tested with flow cytometry after 24 h induction of M1 polarization (20 ng/mL IFNγ + 100 ng/mL LPS), M2 polarization (20 ng/mL IL-4), or PBS (used as a sham operation) (upper). CD45^+^ F4/80^+^ CD11b^+^ cells were gated and examined for CD80 and CD206 expression. Percentages of CD80^+^ (M1 phenotype, down left) cells and CD206^+^ (M2 phenotype, down right) cells within total CD45^+^ F4/80^+^ CD11b^+^ subsets are presented as bar graphs. ** *p* < 0.01, *** *p* < 0.001, Student’s *t*-test, *n* = 3/group, ns—not statistically significant.

**Figure 4 ijms-24-07888-f004:**
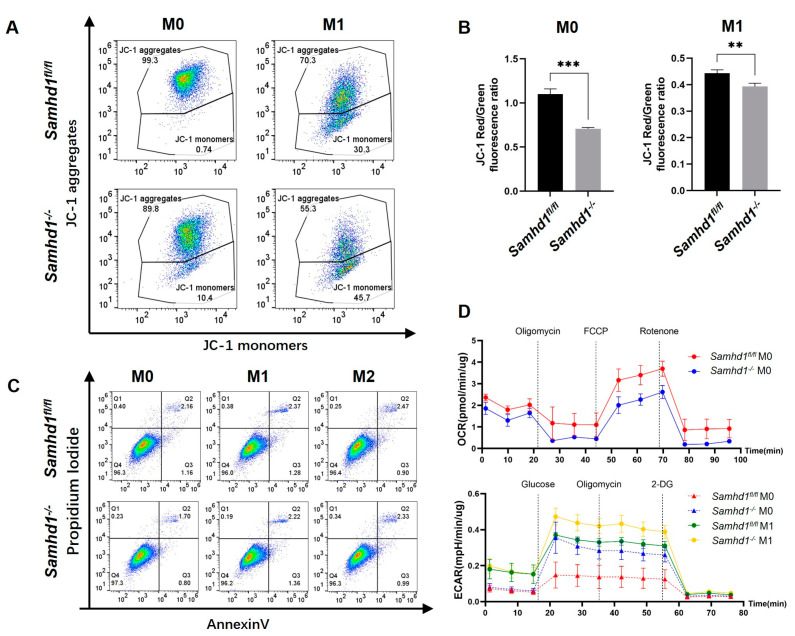
*Samhd1* deficiency impairs mitochondrial membrane potential and function. (**A**,**B**) *Samhd1^−/−^* and control macrophages were labeled with JC-1 after M1-induction for 12 h (PBS was used as a sham operation, represented as M0). (**A**) The proportions of aggregates and monomers of JC-1 in cells were tested with flow cytometry. (**B**) Cell fluorescence intensity in red channel (an Ex wavelength of 525 ± 10 nm and an Em wavelength of 590 ± 10 nm) and green channel (an Ex wavelength of 490 ± 10 nm and an Em wavelength of 535 ± 10 nm) was quantified with a fluorescence plate reader and the ratios of red to green fluorescence intensity are presented as bar graphs. ** *p* < 0.01, *** *p* < 0.001, Student’s *t*-test, *n* = 3/group. (**C**) Representative experiments of *Samhd1^−/−^* and control macrophages in the resting, M1-polarized, and M2-polarized states were tested with flow cytometry for their apoptosis conditions. (**D**) The oxygen consumption rate (OCR) and extracellular acidification rate (ECAR) were detected with mitochondrial stress assay and glycolysis stress assay, respectively, in *Samhd1^−/−^* and control macrophages after 12 h induction of M1 polarization (PBS was used as a sham operation, represented as M0).

**Figure 5 ijms-24-07888-f005:**
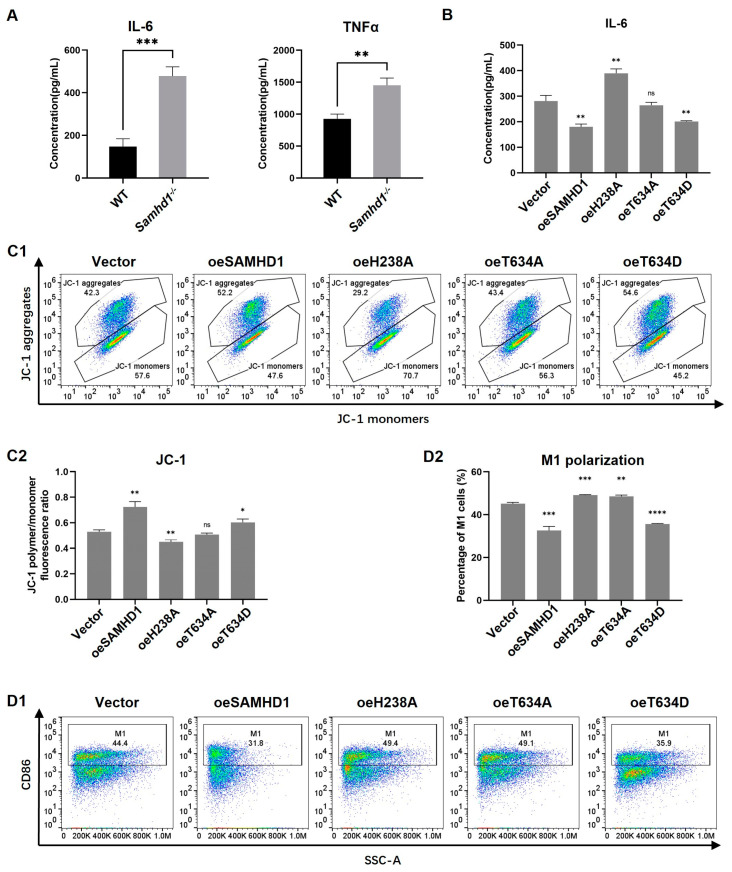
The inhibition of IL-6 and protection of mitochondria were dependent on the dNTPase function of SAMHD1 and regulated by the phosphorylation site T634. (**A**–**D**) *Samhd1^−/−^* and control RAW264.7 cells were established. (**A**) The protein expression of IL-6 and TNFα in cell supernatants was detected after 12 h LPS stimulation (100 ng/mL). (**B**,**C**) *Samhd1^−/−^* RAW264.7 cells were electro-transfected with empty vectors or vectors of full-length SAMHD1 or SAMHD1 mutants such as H238A, T634A, and T634D. After 24 h of transfection, cells were stimulated with LPS (100 ng/mL) for 12 h. (**B**) The protein expression of IL-6 in cell supernatants was detected and is presented as bar graphs. *p* value of vector group relative to paired other groups. ** *p* < 0.01, Student’s *t*-test, *n* = 3/group. (**C1**) Cells were labeled with JC-1, and the proportion of intracellular JC-1 aggregates and monomers was tested with flow cytometry. (**C2**) Cell fluorescence intensity in red channel (an Ex wavelength of 525 ± 10 nm and an Em wavelength of 590 ± 10 nm) and green channel (an Ex wavelength of 490 ± 10 nm and an Em wavelength of 535 ± 10 nm) were quantified with a fluorescence plate reader and the ratios of red to green fluorescence intensity are presented as bar graphs. * *p* < 0.05, ** *p* < 0.01, Student’s *t*-test, *n* = 3/group, ns—not statistically significant. (**D1**) After 24 h of transfection, cells were induced to be M1-polarized (20 ng/mL IFNγ + 100 ng/mL LPS). After another 24 h, M1 polarization of the cells was tested with flow cytometry. Percentage of CD86^+^ cells was quantified as M1-polarized macrophages. (**D2**) The percentage of M1-polarized macrophages was detected and is presented as bar graphs. *p* value of vector group relative to paired other groups. ** *p* < 0.01, *** *p* < 0.001, **** *p* < 0.0001, Student’s *t*-test, *n* = 3/group.

**Figure 6 ijms-24-07888-f006:**
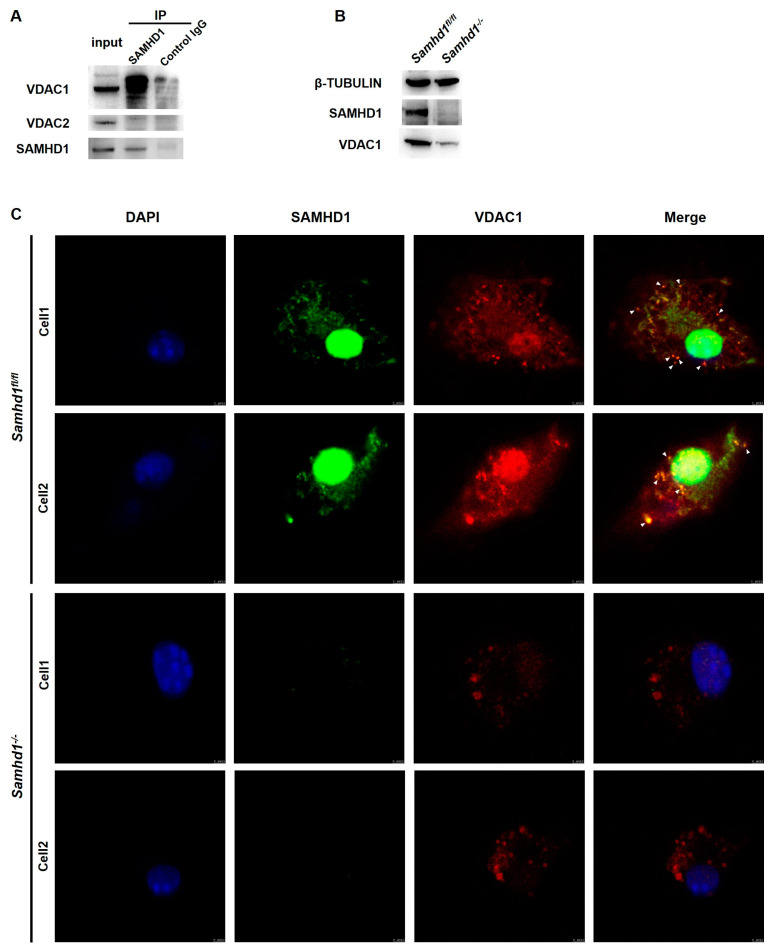
SAMHD1 interacts with VDAC1. (**A**) Immunoblot analysis of VDAC1, VDAC2, and SAMHD1 in anti-SAMHD1 immunoprecipitated lysates of wild-type RAW264.7 cells, with IgG as an isotype control. (**B**) The protein expression of SAMHD1 and VDAC1 was tested with western blotting in *Samhd1^−/−^* and control macrophages. (**C**) Co-localization of SAMHD1 and VDAC1 was observed with immunofluorescence assay under a confocal microscope (Magnification, 200×). Green SAMHD1 interacted with red VDAC1 to form the clustered co-localization sites (labeled with white arrows). The nuclei were shown in blue. The specificity of the SAMHD1 antibody was verified in *Samhd1^−/−^* macrophages.

**Table 1 ijms-24-07888-t001:** Fluorescence signal channels of flow cytometry.

Fluorochrome	Excitation Wavelength (nm)	Emission Wavelength (nm)
JC-1 aggregates	561	566.1–589.5
JC-1 monomers	488	506.1–554.2
AF647	637	657.7–679.9
AF700	637	701.6–744.2
BV421	405	413.6–438
PE	561	566.1–589.5
PI	488	612.6–646.6
FITC	488	506.1–554.2

## Data Availability

The data presented in this study are openly available in raw data at Mendeley Data, https://doi.org/10.17632/6c2sbrfycw.1 (accessed on 15 April 2023).

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
