# Peer review of "SAMHD1 Attenuates Acute Inflammation by Maintaining Mitochondrial Function in Macrophages via Interaction with VDAC1"

_ijms, 2023, doi:10.3390/ijms24097888_

Round 1
Reviewer 1 Report
The authors presented a study that investigates the potential role of SAMHD1 in macrophage polarization and LPS-induced immune response. VDAC1 was identified as a potential interacting protein of SAMHD1. While the results are interesting, there are a few comments to be further addressed.
1. What are the new discoveries in this study compared to the PNAS study (PMID:29610295)?
2. In the LPS-TLR4 pathway, there are a number of feedback molecules identified already, such as A20, IRAK-M, etc. How does SAMHD1 compare with these well-established feedback regulators in terms of functional significance or druggability?
Reviewer 2 Report
The presented work determined the roles of SAMHD1 in LPS-induced inflammatory responses in the Samhd1 gene knockout (Samhd1-/-) mouse strain and RAW264.7 cell line. The authors claimed that SAMHD1 is critical in controlling M1-polarization and maintaining the mitochondrial integrity of macrophages. While the manuscript presents some valuable information, the data should be more thorough to support the authors' main conclusions.
Given the multiple bands near the VDAC1 location in Fig. 6A, the specificity is insufficiently established. Moreover, the interaction between SAMHD1 and VDAC1 (Fig. 6A, although the data is confirmed) and the reduced VDAC1 expression in Samhd1-deficient cells (Fig. 6B) does not indicate a causal relationship between the SAMHD1-VDAC1 physical association and the SAMHD1’s roles in preventing inflammation and M1 polarization of macrophages. The following experiments may clarify this data and further support the conclusion.
1. In Figure 6A, the authors could conduct IP of VDAC2 to detect SAMHD1.
2. In Figure 6A, to better correlate the SAMHD1-VDAC1 interaction with mitochondrial dysfunction, the authors could determine the interaction with and without LPS co-culture of cells. Further, it’s ideal for testing the interaction of VDAC1 with one or more SAMHD1 possessing amino acid substitution listed in Fig. 5C. To confirm the impact of specific amino acid changes on the outcomes, the authors should assess the expression levels of those Samhd1 mutants as well.
3. In Figure 6B, the authors could determine the specificity of VDAC1 suppression. The overall mitochondrial amounts could be decreased in Samhd1-deficient cells.
4. In Figure 6B, it’s worth characterizing the impact of Vdac1 re-expression on Samhd1 knockout RAW264.7 cell. If VDAC1 suppression is vital in displaying Samhd1-deficiency effects, the VDAC1 expression back to the control levels will recover the phenotypes of Samhd1-deficient macrophage.
Reviewer 3 Report
The manuscript by Xu et al. reports the results of their in vivo and in vitro study on the role of the regulatory molecule “SAM and HD domain containing deoxynucleoside triphosphate triphosphohydrolase 1” (SAMHD1) in TLR4-activated inflammation in macrophages. The work extends pervious findings of the group and is of potential interest for the study of inflammatory mechanisms and, in general, of immune function. However, the methodological part of the manuscript is insufficiently described and, thus, cannot be adequately evaluated. In addition, there are defficiencies in experimental design and data presentation and interpretation. All together, these issues prevent acceptation of this manuscript, unless satisfactorily addressed by the authors.
1. Flow Cytometry (FCM) methods:
1.1 General: In general, FCM methods are poorly described. Information on the laser illumination and filter selection for each assay should be included. While there is minimal description of mitochondria (JC-1) and apoptosis (Annexin V-FITC/PI) assays, there is no description at all for the immunophenotypic studies using monoclonal antibodies with macrophages and Raw cells.
1.2 FCM experimental design: Authors do not provide information on the mandatory controls that should be included in the FCM assays of the manuscript. They must provide graphical evidence (by means of appropriate FCM graphs) of the application of viability controls (to exclude dead cells and debris); the staining controls including Fluorescence-minus-One controls, (to establish the negative/positive thresholds for markers that have no on/off expression, as CD80, CD86 and CD206); the compensation of fluorescence; the sequential gating strategy of macrophages based on F480+/CD11b+ subsets. Not having performed such controls may invalidate the specificity of the data obtained and the interpretation of the results, therefore, authors should give proof of their inclusion.
1.3 Data presentation: The names of the axes in some of the FCM dotplots are wrong. They should not be generic names indicating the typical fluorochromes detected by the channel (e.g. FITC for green-fluorescence detector or PE for yellow-fluorescence detector) but the name of the fluorescent marker actually studied in each axis, as follows. Figs. 4A and 5C, X-axis: JC-1 monomers; Y-axis: JC-1 aggregates. Fig. 4C, Y-axis: PI (or, better, Propidium Iodide).
In general, legend to Figures should be revised for being self-informative and coherent with other figures.
1.4 FCM data quantitation: Some important statements are based only on a representative FCM graph, e.g. “As expected, the proportion of M1-polarized lung macrophages was markedly higher in Samhd1-/- mice than that of control mice”, lines 161-162. Also, the statement “Thus, the deficiency of Samhd1 facilitated M1-polarization of macrophages” in lines 167-169 is supported only by the example of Figure 3B, which displays only a 15% difference in M1 and less than 10% in M2 populations, based on the % of cells in both regions provided in the graph.
This, and other statements based on single FCM graphs, should be backed by statistically sound data (e.g. Fig. 3B and 5B). Interestingly, the changes in JC-1 populations exemplified in Fig. 4A (% of cells in JC-1 high- or low-red fluorescence) seem not consistent with the data obtained on the fluorimeter (Fig. 4B).
2. Data interpretation and nomenclature:
JC-1 ratio of red/green fluorescences is a primary indicator of mitochondrial membrane potential, not of mitochondrial “quality” or “integrity”. This should be clearly indicated in the text.
JC-1 red fluorescence is not due to JC-1 polymers, but to JC-1 aggregates. Please correct in several appearances.
The terms “seahorse” mitochondrial stress assay and glycolisis stress assay are incorrect, as “seahorse” is the commercial name of the instrument used to measure oxygen consumption and extracellular pH, in such assays.
3. Experimental design:
Authors should justify why in some experiments only M1 macrophage polarization and in other M1 and M2 polarization is induced.
Round 2
Reviewer 2 Report
The authors addressed most of the comments and suggestions of this reviewer. Nevertheless, the presented data do not establish a causal relationship between SAMDH1-VDAC1 interaction and mitochondrial membrane potential/function maintenance. The authors should soften the conclusion if they cannot provide additional supporting data. In the abstract, the authors should consider “Conclusion: SAMHD1 is critical in controlling TLR4-induced acute inflammation and M1 polarization of macrophages, which may depend on SAMHD1’s interaction with VDAC1. Our data present a novel regulatory mechanism of TLR signaling upon LPS stimulation.”
Author Response
The reviewer made a very pertinent comment for the conclusion of our work. We will revise the abstract as the reviewer suggested.